## [Transparent Peer Review file · Nature Communications]

Covalent Warhead Assembly in Fostriecin Biosynthesis Involves Malonylation-Lactonisation by a Bifunctional Thioesterase and Enzymatic Demalonylation

Corresponding Author: Professor Frank Hahn

Version 0:

Reviewer comments:

Reviewer #1

(Remarks to the Author)

This research article describes on a detailed analysis of the biosynthetic mechanisms for a motif crucial to activity expression within natural product structures.

This study focuses on the α,β -unsaturated δ -lactone structures found in Fostriecin, phoslactomycins, and cytostatins, and employs in vitro enzymatic reactions to analyze their biosynthetic mechanisms in detail.

The reconstruction of PKS modules in vitro and its detailed analysis using enzymatic reactions is a noteworthy contribution.

On the other hand, previous studies have already shown that the α,β -unsaturated δ -lactone structure of these natural products is formed from malonyl ester intermediates. Furthermore, all intermediates and proposed substrates are readily predictable, containing no unexpected or surprising results. This study merely experimentally demonstrated the function of the proposed TE.

While TE catalyzed malonyl esterification of the PK chain on the ACP is indeed unique, in TE-catalyzed malonyl esterification of the PK chain on ACP is indeed specific, but in the biosynthesis of thielavin A (Angew. Chem. Int. Ed. 2024, 63, e202402663 (1 of 7)), it is known that the PK chain initially formed on ACP is first loaded onto the serine residue of the TE. Subsequently, the PK chain formed on ACP a second time captured the first chain bound to the TE by ester exchange. Afterward, the dimerized chain on ACP is transferred to the serine of the TE and finally the chain is released via hydrolysis.

The biosynthetic mechanism is predicted as follows: First, the malonyl-CoA on ACP is transferred to TE, where the serine residue undergoes malonyl esterification. Subsequently, the extended PK chain on ACP captures the malonyl group on TE. The malonyl-esterified PK chain on ACP is then transferred back to the serine residue on TE. Finally, the chain on the serine is released via lactonization.

In this biosynthetic mechanism, substrate recognition differs from the above described pathway due to the distinction between the pk chain synthesized by the enzyme itself and the malonyl group serving as the elongation substrate. However, the catalytic mechanisms of TE in both biosynthetic pathways are similar.

This study actually provides a detailed analysis of what is expected but not yet disclosed, and is insufficient in terms of discovering any new features.

While the detailed in vitro analysis constitutes high-quality research, it is not well-suited for the diverse interests of Nature Communications' readership and should be published in a more chemistry-focused journal.

Based on the above, we do not recommend it to publish in Nature Communications.

Reviewer #2

(Remarks to the Author)

In this manuscript, the authors describe the enzymatic processes for the final steps of fostriecin biosynthesis, a linear polyketide natural product. Specifically, they propose enzymatic mechanisms for the O-malonylation and malonyl elimination steps, which are catalyzed by the FosMod8 thioesterase domain and FosM, respectively. The study is strengthened by the use of chemically synthesized and validated substrate surrogates, supported by convincing NMR and LC-HRMS(/MS) characterization of the in vitro enzymatic reaction products, along with informative mutagenesis experiments. A limitation, however, is the absence of experimental structural data on the proteins, which would further substantiate the proposed mechanisms. Overall, this study advances our understanding of the fostriecin pathway by defining the enzymes responsible for the terminal transformations, identifying their likely mechanisms, and establishing the timing of these late-stage steps. These results should be of interest to polyketide researchers, but the manuscript needs to be improved. My recommendations for revision are outlined below.

Minor:

30: The term “reduced polyketide” is not commonly used in the field. I recommend using “polyketide” instead. (throughout the manuscript)

39: Not all “reduced polyketides” are produced by type I PKSs.

39: “modular PKSs” should be used here instead of “multimodular PKSs”. A multimodular PKS is a PKS protein that consists of more than one extension modules. Some PKSs contain only one module.

46: What is “reductive loop domains”?

82-83: Include FosG in Figure 1b.

125: Fig. 1 a >>> Fig. 1a

238: proposed >>> propose

623: imidazol >>> imidazole

624: imidazol >>> imidazole

759: Geneexpression >>> Gene expression

Major:

116-117: What is the rationale for including FosMod7?

160: Give a detailed explanation of exactly what “FosTEII” is. Provide a summary of Ref 41.

162: Provide a numerical value for “modest boost”.

163: What are the “refined insights”?

188: It is not obvious why a mutation in the TE active site would impair KR activity. Provide a plausible explanation.

218: AlphaFold2 is outdated. Experiment should be performed using AlphaFold4.

219-220: This statement is highly speculative. Are there examples where AlphaFold predicts a different conformation for each monomer in a homodimeric protein? Do you get the same result with AlphaFold 4?

228-233: Provide experimental details of the docking experiment.

265: “this mutation significantly compromises the integrity of the module” – this is straight forward to test (e.g. using a thermal shift assay).

270-272: If hydrophobic interaction between the two Leu residues stabilize the active site, why is only lactonization restored and not malonylation?

281: The authors tested whether the TE domain acts on the ACP-bound substrate or on the released elongation product.

What is the significance of this experiment? Why did you suspect the TE to act on the released product? Is there a precedent for such a case? Modular PKSs act in a modular manner. All domains act on the substrate that is attached to the ACP of that module. The authors conclude that O-malonylation occurs on the ACP-bound intermediate. I agree but why was this in question in the first place?

309: Conversion of 25 to 27 is not observed. The figure (Fig. 5c right) should be modified to reflect this (e.g. put an X over the arrow).

326: Provide a numerical value for “slightly increased”.

353-355: Is it possible that phosphorylation takes place on the polyketide chain bound to FosMod8, before it is released?

373: Discussion section is weak in that it simply summarizes the results section. A discussion about future direction, still unclear mechanisms, and engineering is desirable.

745: What experiments were performed using holo-FosACP7 and holo-FosACP8?

757: What experiments were performed using PiKTE?

Figure 4b,c: It's not clear where the entrance and exit are. Add an arrow to clarify this. Label the residues highlighted in purple. What do the different colors (pocket) represent?

Supplementary Figure 29: Include more proteins for comparison. Three is too small for this type of analysis.

Supplementary Figure 39: Please check the correctness of the masses. Top left box shows 531.1952 for calculated and 531.1966 for found. Bottom right box shows 521.1952 for found.

Supplementary Figure 67: These gel pictures are too small. The MW standards must be labeled.

Version 1:

Reviewer comments:

Reviewer #2

(Remarks to the Author)

My initial concerns were sufficiently addressed in the revision. Please fix the tilted carbonyl (highlighted in purple) for

compound 13 in Figure 1. Some residue numbers don't match in the left and right panels of Figure 4.

Reviewer #1 (Remarks to the Author):

This research article describes on a detailed analysis of the biosynthetic mechanisms for a motif crucial to activity expression within natural product structures. This study focuses on the α,β -unsaturated δ -lactone structures found in Fostriecin, phoslactomycins, and cytostatins, and employs in vitro enzymatic reactions to analyse their biosynthetic mechanisms in detail. The reconstruction of PKS modules in vitro and its detailed analysis using enzymatic reactions is a noteworthy contribution.

On the other hand, previous studies have already shown that the α,β -unsaturated δ -lactone structure of these natural products is formed from malonyl ester intermediates. Furthermore, all intermediates and proposed substrates are readily predictable, containing no unexpected or surprising results. This study merely experimentally demonstrated the function of the proposed TE.

While TE catalysed malonyl esterification of the PK chain on the ACP is indeed unique, in TE-catalysed malonyl esterification of the PK chain on ACP is indeed specific, but in the biosynthesis of thielavin A (Angew. Chem. Int. Ed. 2024, 63, e202402663 (1 of 7)), it is known that the PK chain initially formed on ACP is first loaded onto the serine residue of the TE. Subsequently, the PK chain formed on ACP a second time captured the first chain bound to the TE by ester exchange. Afterward, the dimerised chain on ACP is transferred to the serine of the TE and finally the chain is released via hydrolysis.

The biosynthetic mechanism is predicted as follows: First, the malonyl-CoA on ACP is transferred to TE, where the serine residue undergoes malonyl esterification. Subsequently, the extended PK chain on ACP captures the malonyl group on TE. The malonyl-esterified PK chain on ACP is then transferred back to the serine residue on TE. Finally, the chain on the serine is released via lactonization.

In this biosynthetic mechanism, substrate recognition differs from the above described pathway due to the distinction between the pk chain synthesised by the enzyme itself and the malonyl group serving as the elongation substrate. However, the catalytic mechanisms of TE in both biosynthetic pathways are similar.

This study actually provides a detailed analysis of what is expected but not yet disclosed, and is insufficient in terms of discovering any new features.

While the detailed in vitro analysis constitutes high-quality research, it is not well-suited for the diverse interests of Nature Communications' readership and should be published in a more chemistry-focused journal.

Based on the above, we do not recommend it to publish in Nature Communications.

We appreciate the reviewer's feedback. The comments have provided us with a clearer understanding that the relevance of our work needs to be more effectively presented. We have amended the manuscript accordingly. However, we respectfully disagree with the assessment that our findings merely confirm an obvious mechanism and that the reactions catalysed by FosTE simply resemble known processes for the following reasons:

The reviewer's summary of our proposed mechanism contains an important inaccuracy. The O-malonylation does **not** proceed via a malonyl-ACP intermediate that is formed from malonyl-CoA and ACP-SH beforehand (e. g. by an AT domain). Instead, FosTE directly self-acylates using malonyl-CoA, which is a unique mechanism for malonyl introduction into an assembly line (Fig. 5a). The ACP-domain is not required for this activity. We have demonstrated this by assays of standalone FosTE with compound **31** (Fig. 5c) and by showing that FosTE self-malonylates in the presence of malonyl-CoA (see Supplementary Fig. 67).

FosTE exhibits a novel catalytic sequence unprecedented for modular PKS TE domains. To our knowledge, no prior examples exist in which a TE domain from a type I PKS accepts both, a CoA-bound extender unit as well as an ACP-bound polyketide chain as substrates and catalyses multiple different acyl migrations. FosTE represents the first enzyme for which the following catalytic sequence has been conclusively demonstrated in vitro: (1) self-acylation with malonyl-CoA, (2) transacylation of the malonyl group onto an ACP-bound polyketide intermediate, (3) transfer of the O-malonylated intermediate from the ACP to the TE catalytic serine, and (4) δ -lactonisation. This mechanism has been elucidated for the first time for the pharmacologically important class of α,β -unsaturated δ -lactone (AUDL) natural products, for which no prior mechanistic evidence existed.

Although other TEs share some of these features, none displays the complete set characteristic of FosTE. TE-mediated alcohol esterifications in the biosyntheses of FR900359, necroxime A, and salinamide require prior loading of the acyl residue onto a carrier protein, followed by transacylation onto macrocyclic polyketides/peptides (refs. 40–42). These enzymes lack the multifunctionality of FosTE and perform acylation on a free substrate rather than on a PKS intermediate. The TE domains of the conglobactin, elaiophylin, and thielavin A-forming PKSs catalyse multiple inline acylations of secondary hydroxyls of PKS intermediates (refs. 44–46; see next paragraph). They however remain restricted to transferring acyl groups supplied by their own assembly line and do not introduce molecular entities from outside. TE_B domains catalyse inline backbone hydroxyl acylation using CoA thioesters, but are monofunctional. None of these types of TEs processes malonyl units.

The cited precedent (Thielavin A) differs fundamentally from our system. We thank the reviewer for bringing this example, which we had previously overlooked, to our attention. ThiA is a monomodular PKS with an iteratively-acting TE domain and an unusual domain architecture. The ThiA-TE assembles monomeric units into a trimer before final hydrolysis. During this, ThiA-TE exclusively acylates itself with ACP-bound polyketide intermediates processed by upstream domains (as is conventional). It then utilises a hydroxyl of the phenolic intermediate bound to the ACP immediately upstream as the nucleophile for release from the catalytic serine. In contrast to FosTE, which processes malonyl-CoA and a complex ACP-acyl, the acyl units transferred by ThiA-TE are ACP-bound mono- or dimers of methylorsellinic acids and thus show clear similarities to each other. This mechanistic principle in thielavin A biosynthesis resembles that described for the Ela-TE and Cong-TE, which we have presented in the introduction (refs. 44–46). This information on ThiA was added to the manuscript.

The TE domain was not an evident or obvious candidate for this activity. While TE-mediated catalysis may seem plausible in retrospect, this plausibility does not constitute proof but places it among a number of other proposals. Alternative mechanisms, for example involving other FosMod8 domains required systematic consideration.

There are numerous reports suggesting that certain PKS domains and their homologs can perform oxygen ester formation and/or malonyl group transfer. Being the canonical malonyl introducer in modular type I PKS systems, the AT domain would have been the most intuitive candidate to us. The hypothesis that a non-canonical AT domain activity might be responsible, was indeed initially postulated by Reynolds *et al.* in their pioneering description of the malonylated lactone in phoslactomycin biosynthesis (ref. 25, revised version of the manuscript). As mentioned in the manuscript's introduction, ATs are known to catalyse the transfer of malonyl groups to secondary alcohols in substrates (refs. 33–35) or the esterification of polyketide backbone hydroxyls (refs. 37–38). We have added a sentence to the introduction that highlights the relevant hypothesis of *trans*-malonylation by an AT encoded outside the *fos* gene cluster. This is in general well documented for ATs (ref. 73), evidenced by the widespread contribution of FAS malonyl-CoA:ACP ATs to priming modular and type II PKSs (refs. 29–31).

It is described that KSs can be capable of malonyl transfer and ester formation as proven for CerJ, which transfers a malonyl unit to a secondary alcohol in cervimycin biosynthesis (ref. 36), as well as NonJ and NonK, which esterify polyketide backbone hydroxyls for macrotetrolide-formation in nonactin biosynthesis (ref. 39). Self-malonylation is a commonly observed feature of type II PKS-ACPs. Even though this reaction does not involve a small molecule or ACP-bound acceptor, it shows that ACPs can bind malonyl-CoA and force malonyl exchange reactions with nucleophiles in their active centre (see ref. 32).

By contrast, no TE domains are known to catalyse malonyl transfer. At the outset of our study, the likelihood of a C-terminal PKS-TE domain accomplishing this starting from malonyl-CoA was deemed low, given that primary and tertiary structure-based classification of TEs into families strictly goes along with CoA or ACP thiol preference (ref. 47). C-terminal TE domains of PKS assembly lines are in families TE15–17 and act on ACP-bound thioesters (FosTE is in family TE17). We are not aware of any TE15–17 member that uses a CoA thioester as a substrate. TE_b domains are centrally integrated in assembly lines and catalyse esterification using acyl-CoAs. They are however monofunctional and do not accept malonyl or ACP thioesters (ref. 43).

The scope of the manuscript is not restricted to FosTE catalysis. Our work provides further critical progress for understanding fostriecin pharmacophore (AUDL and phosphate ester) biosynthesis, such as the first in vitro activity test of a demalonylating enzyme (FosM) and a full process reconstitution. It reveals the precise timing of all pharmacophore-forming steps, providing information on enzyme specificities that is crucial for successful engineering of fostriecin biosynthesis and chemoenzymatic synthesis strategies. It would for example be synthetically elegant and efficient to introduce the complete AUDL enzymatically at a late stage of a synthesis using the presented enzymes. In reported fostriecin total syntheses, the AUDL is introduced via building blocks prepared in 4–8 steps (ref. 14). Its sensitivity further complicates synthetic strategies, increasing the overall step count. Findings such as our observation that early FosH-catalysed phosphorylation prevents processing by FosMod8, together with the successful full reconstitution (see Figs. 2 and 6), provide a clear framework for designing an effective multienzymatic system.

Finally, we would like to emphasise that, comparable in vitro studies on late-stage modular PKS processes remain extremely rare, with most reports deriving from the foundational work of Sherman *et al.* on macrolactone biosynthesis (see e.g. ref. 54 and related work over the last 20 years). In our work, we not only developed optimised transformations of active PKS modules, but also synthesised numerous complex substrate surrogates to enable sophisticated in vitro assays. The considerable challenges of this approach further underscore the significance of the achieved results.

Reviewer #2 (Remarks to the Author):

Minor:

30: The term “reduced polyketide” is not commonly used in the field. I recommend using “polyketide” instead. (throughout the manuscript)

39: Not all “reduced polyketides” are produced by type I PKSs.

39: “modular PKSs” should be used here instead of “multimodular PKSs”. A multimodular PKS is a PKS protein that consists of more than one extension modules. Some PKSs contain only one module.

To avoid a lengthy description of the diverse PKS subtypes, we originally intended to straightforwardly guide the reader towards the relevant type of PKS: bacterial PKS systems, which consist of multiple modules with reducing activity. While iteratively-acting PKS modules with reducing activity are common in fungi, they are exceptions from the rule in bacteria. Iterative systems in bacteria are predominantly type II PKS, which assemble aromatic products. We therefore used the common term ‘multimodular PKS’. The short form ‘reduced polyketide’ was used to differentiate the products from these resulting from PKS systems devoid of reducing activity. We have adjusted the text in this section of the introduction, which will hopefully adequately address the reviewer's objections without extending the introduction unnecessarily.

46: What is “reductive loop domains”?

‘Domains’ needed to be removed here. We have added a sentence to explain how the individual composition of KR, DH and ER domains in reductive loops of PKSs influences the structure of their products.

82-83: Include FosG in Figure 1b

125: Fig. 1 a >>> Fig. 1a

238: proposed >>> propose

623: imidazol >>> imidazole

624: imidazol >>> imidazole

759: Geneexpression >>> Gene expression

These changes have been made.

Major:

116-117: What is the rationale for including FosMod7?

Neither FosMod7 nor any of its homologous modules (such as PlmMod6 or Pn6) have previously been successfully reconstituted or characterised *in vitro* with respect to their enzymatic activity. As examples of crosstalk between domains of different PKS modules (ref. 48) or reports on trans-acting AT enzymes (ref. 28) are literature-described and the process of malonyl introduction was cryptic, we saw it as a realistic scenario that FosMod7 is a potential contributor to this non-canonical process. To unambiguously rule out this possibility, FosMod7 was therefore included in the investigation from the outset. Motivated by the comment from reviewer 1, we have now additionally mentioned the possibility that *trans*-acting enzymes contribute to O-malonylation more prominently in the introduction.

160: Give a detailed explanation of exactly what “FosTEII” is. Provide a summary of Ref 41.

We have added a brief summary of ref. 41 (now ref. 53) along with an explanation for the choice of FosTEII. It had previously proven useful to improve in vitro productivity of the *pre*-phoslactomycin-producing Pn-PKS.

162: Provide a numerical value for “modest boost”.

163: What are the “refined insights”?

We now present the maximum conversion increases achieved by adding FosTEII to the in vitro assays with the individual PKS modules.

We have reworded the sentence regarding 'refined insights' to make our statement clearer: The increases in conversion achieved through improved protein purification and the addition of FosTEII enabled detailed examination of the complex product profile by LC-MS.

188: It is not obvious why a mutation in the TE active site would impair KR activity. Provide a plausible explanation.

The observed loss or reduction of β -ketoreductase activity in the mutant may be indicative of a dynamic coupling between the β -ketoreduction step and the transient O-malonylation of the β -hydroxy intermediate by the TE domain. It is conceivable that, in the wild-type system, the ketoreduction occurs only after or concurrently with the self-O-malonylation of the TE catalytic serine. In the catalytic serine mutant, the inability of the TE domain to undergo self-malonylation could disrupt the conformational coupling or interdomain communication within the module, thereby slowing or preventing the β -ketoreduction. As a consequence, the β -keto intermediate tethered to the ACP might accumulate in a stalled state and be prematurely hydrolysed or released before further processing can take place. Previous studies on late modules of Ery-PKS and Pik-PKS, which used complex PKS intermediate surrogates, have shown that changes in the interaction between the substrate and the PKS domains have subtle effects on the dynamics between KR and TE (see e.g. ref. 54). These changes can result in the formation of non-reduced by-products. A passage explaining this has been added to the Results section of the manuscript and it has also been included in the Discussion.

218: AlphaFold2 is outdated. Experiment should be performed using AlphaFold4.

We thank the reviewer for their comment and for emphasising the importance of using the most current structural prediction tools. As of December 2025, AlphaFold 4 has not been officially released or made available to the public by DeepMind. While references to “AlphaFold 4” appear in informal online sources, these do not correspond to any verified or publicly accessible version. The latest publicly available version is AlphaFold 3, which is accessible through the official AlphaFold 3 server. We believe the reviewer may have been referring to this version. Accordingly, we have updated all structures previously generated with AlphaFold 2 to the corresponding AlphaFold 3 models (Fig. 4; Supplementary Figs. 24–28). The docking studies based on the FosTE AlphaFold 3 models, predicted with high confidence, were repeated as shown in Fig. 4, and the text has been revised accordingly to reflect these updated results.

219-220: This statement is highly speculative. Are there examples where AlphaFold predicts a different conformation for each monomer in a homodimeric protein? Do you get the same result with AlphaFold 4?

We appreciate the reviewer's comment and acknowledge that this aspect of our original interpretation was somewhat speculative. This explicit case was only reported in one additional publication (<https://doi.org/10.1016/j.omtn.2024.102284>). As noted in the previous point, we have now updated both the structural models and the docking analysis using AlphaFold 3.

In the AlphaFold 2 predictions, we obtained five models. One of these five models showed an explicitly asymmetric homodimer, in which the two monomers displayed different active-site cavity geometries. The remaining models were symmetric with respect to the two monomers but differed from each other in the size and shape of this cavity. In other words, they sampled the same range of pocket geometries that appeared in the asymmetric outlier model, but without breaking homodimer symmetry.

After re-analysis with AlphaFold 3, we again observed two plausible arrangements of the active-site region. Specifically, among the five highest-confidence AlphaFold 3 models, three exhibited an expanded cavity and two showed a more restricted cavity. However, in contrast to the AlphaFold 2 outlier, all corresponding AlphaFold3 models were symmetric with respect to the homodimer, making the questioned statement above obsolete.

There is published work showing that AlphaFold can sample alternative structural states for the same protein sequence, in some cases corresponding to experimentally observed functional states (<https://doi.org/10.1021/acs.jcim.5c00906>; <https://doi.org/10.1371/journal.pcbi.1010483>; <https://doi.org/10.7554/eLife.75751>; <https://doi.org/10.1002/pro.4368>). Accordingly, we have revised the manuscript. The discussion now focuses instead on possible ligand-dependent differences/states in cavity volume suggested by the AlphaFold 3 models, together with the updated docking results, reflecting a more cautious interpretation aligned with current literature.

228-233: Provide experimental details of the docking experiment.

The given experimental details in the Methods section have been adjusted for model generation by AlphaFold 3 and subsequent docking.

265: "this mutation significantly compromises the integrity of the module" – this is straight forward to test (e.g. using a thermal shift assay).

We were referring to the catalytic integrity of the module rather than its structural integrity. The wording in the manuscript has been amended to reflect this.

We appreciate the reviewer's suggestion to perform a thermal shift assay to assess global stability. However, in this case we believe that additional stability measurements would not change the interpretation and are outside the main scope of this work. The R198L mutant is expressed in soluble form, can be purified under the same conditions as the wild type, and shows identical ion exchange chromatography profiles, supporting that overall folding and surface charge are retained.

Functionally, the mutant specifically loses elongation activity while retaining thioester hydrolysis activity, demonstrating a loss of specific catalytic function rather than global misfolding or collapse. Because our focus here is on the functional consequence of this single amino acid substitution, and because this functional phenotype is already directly demonstrated by the activity assays using the same substrate as for the wild type, we do not expect an additional thermal shift assay (which primarily

reports global melting behavior rather than catalytic competence) to provide crucial insight in the context of this study.

270-272: If hydrophobic interaction between the two Leu residues stabilize the active site, why is only lactonization restored and not malonylation?

The hydrophobic interaction between the two leucine residues primarily stabilises the overall architecture of the active site, likely facilitating the correct positioning necessary for lactonisation to occur efficiently. However, transmalonylation requires specific polar/electrostatic interactions to correctly position and activate the malonyl moiety within the active site. Our docking studies indicate that conserved arginine residues form such critical polar interactions with the malonyl group, likely facilitating the transition state stabilisation and binding of the O-malonylated intermediate to the thioesterase domain. Therefore, while the Leu-Leu hydrophobic interaction restores the structural integrity needed for lactonisation, the absence or impairment of polar interactions, particularly involving arginine residues, prevents the restoration of malonylation activity.

281: The authors tested whether the TE domain acts on the ACP-bound substrate or on the released elongation product. What is the significance of this experiment? Why did you suspect the TE to act on the released product? Is there a precedent for such a case? Modular PKSs act in a modular manner. All domains act on the substrate that is attached to the ACP of that module. The authors conclude that O-malonylation occurs on the ACP-bound intermediate. I agree but why was this in question in the first place?

These experiments were carried out to distinguish between two previously described scenarios for backbone acylation. As stated in the introduction, C-terminal TEs from PKSs involved in FR900359, necroxime A and salinamide biosynthesis have been shown to acylate mature macrocycles that are not attached anymore to assembly lines (refs. 40–42). On the other hand, inline O-acylation, catalysed by the C-terminal PKS-TE, occurs in the biosynthetic pathways of conglobactin, elaiophylin and thielavin A (refs. 44–46). A sentence explaining this has been added to the manuscript (lines 318–319).

309: Conversion of 25 to 27 is not observed. The figure (Fig. 5c right) should be modified to reflect this (e.g. put an X over the arrow).

The arrow was modified.

326: Provide a numerical value for “slightly increased”.

The value has been added.

353-355: Is it possible that phosphorylation takes place on the polyketide chain bound to FosMod8, before it is released?

For this, we would like to refer the reviewer to the FosH assays and the full reconstitution experiments shown in Fig. 6., Supplementary Figs. 20 and 21 as well as the accompanying discussion described in lines 387-400 (revised version).

Although otherwise showing relaxed substrate specificity, FosH only phosphorylated the close surrogate of the FosMod7 precursor (**16**) to a low degree (~10%), making its action before this point of PKS processing unlikely. Although FosMod8 precursor surrogate **17** was phosphorylated to a higher degree by FosH, the resulting intermediate **37** was not at all accepted by FosMod8 (Supplementary Fig. 20, panel 2 and Supplementary Fig. 21), showing that phosphorylation is detrimental to late-stage PKS processing. The surrogate of the immediate FosTE precursor (**38**) was only phosphorylated by FosH to a low degree. Although the lack of ACP-binding mimic makes this surrogate a less realistic one than **16**, **17** or the ACP-bound analog of **38** (formed upon elongation and reduction of **17** by FosMod8), it clearly points towards the same direction.

From this we conclude that phosphorylation during biosynthesis only take place after release from the assembly line.

373: Discussion section is weak in that it simply summarizes the results section. A discussion about future direction, still unclear mechanisms, and engineering is desirable.

We agree with this statement of the reviewer and have added these points along with some others to the discussion section.

745: What experiments were performed using holo-FosACP7 and holo-FosACP8?

These were control experiments to proof complete *in vivo* 4'-phosphopantetheinylation of these carrier proteins in *E. coli* BAP1. The respective LC-HRMS chromatograms are shown in Supplementary Figs. 22 and 23.

757: What experiments were performed using PikTE?

PikTE was used in the synthesis of FosMod7 product surrogate **19** by hydrolysis of SNAC thioester **17**. The procedure is shown in the section 'Synthesis of substrate surrogates and reference compounds' in the Supporting Information.

Figure 4b,c: It's not clear where the entrance and exit are. Add an arrow to clarify this. Label the residues highlighted in purple. What do the different colors (pocket) represent?

An arrow has been added to the newly generated α Fold3 models, which are now shown in Fig. 4. The different colours represent hydrophobicity profiles, which are explained in the Figure caption: from red areas representing hydrophobic regions to blue areas representing hydrophilic regions.

Supplementary Figure 29: Include more proteins for comparison. Three is too small for this type of analysis.

Supplementary Fig. 29 was intended to provide an overview of the non-canonical FosTE homologs in a condensed form. We have now added the two homologs of PlmTE from the phoslactomycin-producing *S. auratus* and *S. platensis* (PnTE) as the only other available TE domains of homologous PKSs with a supposedly similarly acting TE domain as FosTE to the alignment. A multiple sequence alignment of altogether 13 TE domains is shown in Supplementary Fig. 30 to provide a broad basis for comparison.

Supplementary Figure 39: Please check the correctness of the masses. Top left box shows 531.1952 for calculated and 531.1966 for found. Bottom right box shows 521.1952 for found.

This has been corrected.

Supplementary Figure 67: These gel pictures are too small. The MW standards must be labeled.

These changes have been made (Supplementary Fig. 68 and subsequent).

Reviewer #2 (Remarks to the Author):

My initial concerns were sufficiently addressed in the revision. Please fix the tilted carbonyl (highlighted in purple) for compound 13 in Figure 1. Some residue numbers don't match in the left and right panels of Figure 4.

These changes have been made.